# Resilience and Strategic Emotional Intelligence as Mediators between the Disconnection and Rejection Domain and Negative Parenting among Female Intimate Partner Violence Victims

**DOI:** 10.3390/brainsci13091290

**Published:** 2023-09-06

**Authors:** Klaudia Sójta, Małgorzata Juraś-Darowny, Aleksandra Margulska, Wioletta Jóźwiak-Majchrzak, Anna Grażka, Dominik Strzelecki

**Affiliations:** 1Department of Affective and Psychotic Disorders, Medical University of Lodz, Czechoslowacka Street 8/10, 92-216 Lodz, Poland; klaudia.krakus@stud.umed.lodz.pl (K.S.); anna.bartosiewicz@stud.umed.lodz.pl (A.G.); 2Institute of Psychology, Faculty of Educational Sciences, University of Lodz, Rodziny Scheiblerów 2, 90-128 Lodz, Poland; malgorzata.juras.darowny@edu.uni.lodz.pl; 3Department of Adolescent Psychiatry, Medical University of Lodz, Czechoslowacka Street 8/10, 92-216 Lodz, Poland; aleksandra.margulska@umed.lodz.pl; 4Department of Applied Sociology and Social Work, University of Lodz, Rewolucji 1905 41/43, 90-214 Lodz, Poland; wioletta.jozwiak.majchrzak@edu.uni.lodz.pl

**Keywords:** intimate partner violence, parenting, early maladaptive schemas, resilience, strategic emotional intelligence

## Abstract

(1) Background: The exposure of children to intimate partner violence (IPV) is associated with a wide range of negative effects on children’s development, where as parenting practice is considered to be one of the key factors mediating and mitigating this. Studies have found mixed results regarding the impact of female IPV victimization on maternal parenting practice; however, the most frequently tested hypothesis suggests that the cumulative stress of the IPV experience may emotionally deregulate the mother, contributing to an increased risk of neglected and abusive parenting practices. Little is still known about the factors determining the observed differences in maternal parenting practices among IPV victims. Thus, in our study, we use mediation models to provide preliminary results exploring the role of resilience and strategic emotional intelligence in the relationship between women’s disconnection and rejection (D/R) schema domain and maternal parenting practice among IPV victims. (2) Methods: A total of 48 female survivors of IPV and 48 age-matched women with no prior experience of IPV completed a set of tests examining parenting practices, the D/R domain, resilience and emotional intelligence. (3) Results: IPV victimization was associated with significantly higher rates of negative parenting practices. The D/R domain was found to be a significant predictor of parental autonomy attitude and level of parental competence, and these relationships were fully mediated by resilience with strategic emotional intelligence and resilience, respectively. (4) Conclusions: The results shed light on the under-researched relationship between early maladaptive schemas and parenting behavior in the context of IPV. The implications for clinical practice and further research can be drawn based on the study findings.

## 1. Introduction

Intimate partner violence (IPV) is a devastating phenomenon with damaging consequences that extend far beyond the individual level [1]. According to estimates by the United Nations Children’s Fund (UNICEF), 133–275 million children are exposed to IPV each year [2]. Exposure to IPV refers not only to the situation where the child is a direct witness to violence between the parents but also to a child’s awareness that any type of violence is present in the parental relationship [3]. Extensive research has demonstrated a broad spectrum of short- and long-term multidimensional effects of IPV exposure on children’s development and well-being [4]. This includes health outcomes, such as increased rate of emergency room visits, asthma and respiratory problems, higher BMI scores and more reported health problems, and deterioration in social and emotional competence, with higher levels of emotional deregulation, increased risk of bullying perpetration and victimization, lower self-efficacy, higher social anxiety and externalizing and internalizing problems [5]. Being exposed to IPV during childhood or adolescence can lead to severe, negative outcomes in adulthood, which include a higher risk of affective and stress-related disorders, violence victimization or perpetration, antisocial behaviors and substance abuse [6]. The abovementioned consequences of children’s exposure to IPV are the result of both the direct effect of witnessing violence and the indirect effect related to the negative impact of IPV on caregivers and their parenting practices [7]. Therefore, parental practice is considered a crucial factor mediating and moderating the link between IPV exposure and children’s outcomes [8].

In the context of the family system, male violence against women is by far more common, hence, in the light of numerous studies, maternal parenting practice is associated with mediating the effect of the consequences of IPV exposure on children’s well-being [9]. Research has revealed inconsistent findings regarding the effects of female IPV victimization on maternal parenting practice. A recent meta-analysis provided evidence of a negative relationship between IPV victimization and secure mother–child attachment, with the relationship increasing accordingly with a lower child age [10]. A number of studies have shown a significantly higher risk of negative parenting behavior among victims of IPV [11,12,13]. Mothers in families where IPV was present were more likely to engage in harsh parental behavior towards their children [14,15]. To explain this relationship, some researchers have used the spillover theory [16]. The spillover hypothesis suggests that negative experiences in one relationship may adversely affect other family relationships, so the cumulative stress of the IPV experience may emotionally deregulate the mother, contributing to an increased risk of neglecting and abusing parenting practices [17]. This may be supported in part by studies of physiological changes in women experiencing IPV, which indicate that IPV victimization affects cortisol diurnal rhythms by increasing evening/bedtime cortisol levels [18] or lowering the level of cortisol while awake in connection with post-traumatic stress disorder(PTSD) [19]. IPV exposure was indirectly associated with child waking cortisol levels, as mothers’ ability to interact positively and warmly with their children was impaired [20]. Another pathway between IPV and maternal parenting practice may involve maternal mental health. Research has clearly established that IPV victimization is associated with a higher risk of developing mental health problems such as depression or PTSD [21]. A growing body of evidence has shown a significant link between maternal PTSD and children’s psychological well-being, specifically externalizing and internalizing behaviors [22,23]. Research by Levendosky et al. showed that the poorer psychological functioning of mothers who experienced IPV was associated with lower parental effectiveness and more insecure attachment to children [24,25]. Nevertheless, the results did not support the hypothesis of a direct negative impact of IPV victimization on parenting behavior [24]. Substantially, a positive effect of experiencing IPV on maternal parenting practice has been documented in other investigations [26,27]. One proposed explanation for these unexpected results is that mothers experiencing IPV attempt to compensate their children for a difficult and stressful family situation through more involved, warm and accepting parenting [24]. However, little is known about the factors determining the observed differences in maternal parenting practice among IPV victims [28].

Pilkington et al., in a meta-analysis of the associations between early maladaptive schemas (EMS), considered as modifiable ontogenetic factor, and IPV victimization and perpetration, documented emerging evidence of an association between IPV victimization and the disconnection and rejection (D/R) schema domain [29]. According to Young’s schema theory, the D/R domain is the resultant of the deprivation of the basic emotional needs of a secure, stable, accepting, warm and predictable relationship with caregivers [30] and is also closely related to the experience of neglect and abuse in the family environment [31]. Although the evidence base is still limited, the results consistently show that parents who suffer from the severity of the D/R domain are more likely to fail to respond effectively to children’s basic emotional needs, i.e., those that are necessary for harmonious development, which contributes to the intergenerational transmission of harsh and hostile or aggressive parenting [32]. These findings are also in line with the results of a meta-analysis by Savage et al., which showed a consistent relationship between women’s experiences of being victimized in childhood and their subsequent negative parenting behaviors towards their children [33].

As presented, female IPV victimization, through various pathways, may impair a mother’s ability to interact warmly and sensitively with her children and poses a certain risk for the mother to engage in negative parenting practice. Given the paramount importance of parenting rearing styles to ensure optimal development of children [34], it is of particular importance to explore the adaptive aspects of parental behavior in the specific context of IPV victimization. Moreover, a recent meta-analysis highlighted the urgent need for more comprehensive parenting theories that would cover both the traumatic aftermath of IPV experiences and a variety of resilience strategies [35]. Resilience is a multidimensional, dynamic process that involves an individual’s ability to adapt healthily and positively during and after exposure to a highly adverse event. It emerges from the complex interactions between various factors, including social, community, familial and individual, throughout life [36]. In the context of parenthood, resilience is conceptualized as the ability to provide competent and high-quality parenting to children, even in the face of adverse personal, familial and social circumstances [37]. It is closely related to the readiness to actively use personal resources, such as skills, abilities and knowledge, in order to effectively and adaptively face challenges and obstacles encountered in the field of child upbringing. Whereas it intuitively appears that emotional intelligence plays a significant role in resilience skills, there is still a limited but consistent evidence base to support this association [38,39,40]. Recent research has shown that higher emotional intelligence in parents is associated with a more authoritative parenting style and that through this relationship children’s aggressiveness is mitigated [41]. Lower levels of emotional intelligence were also directly related to the negative parenting behaviors of mothers in Lee and Kim’s study [42].

Considering the above, in our study we use mediation models to explore the role of resilience and strategic emotional intelligence in the relationship between women’s D/R schema domain and maternal parenting practices in a group of women who experienced IPV victimization. We hypothesized that (1) women who experienced IPV would score significantly higher on the scales of unfavorable parenting practices and significantly lower on the scale of parental competences, (2) the severity of the D/R domain would be associated with unfavorable parenting practices and parental competence, with positive and negative correlations, respectively, and (3) resilience and strategic emotional intelligence would mediate the link between the maternal D/R schema domain and unfavorable parenting practices and parental competence. The results of this study allow us to determine how the IPV experience affects psychological well-being and parenting practice and to assess the protective role of individual resources in this association.

## 2. Materials and Methods

### 2.1. Participants

Initially, 112 women participated in the study. During the study, 10 participants in the study group discontinued the tests. Additionally, 6 sets of questionnaires were excluded due to incomplete data. The declared reasons for withdrawal were feelings related to the YSQ and difficulties with the Emotional Understanding Test (TRE). The final sample for this study consisted of 96 mothers divided into two groups: the study group included 48 women who had experienced IPV in the last year and the comparison group included 48 women who had not previously experienced IPV. The recruitment of study participants took place predominantly in the Łódź Voivodeship. The groups were homogeneous in terms of age. Each participant had full parental rights to at least one child. The women in the study group and the control group were between 20 and 55 years old. The mean age of mothers in the study group was 32.91 ± 7.79, whereas in the control group it was 34.27 ± 2.90. Most participants from the IPV group had a lower educational status (77.08%), were single, divorced or in informal relationships and lived in a large city (64.58%). In turn, the reference group consisted mainly of women with higher education (87.55%) who were married (81.25%) and lived in a big city (56.25%). In the IPV group, 58.33% of women experienced physical or emotional violence in childhood, whereas only 2.08% of women in the non-IPV group reported such experiences. Witnessing childhood domestic violence was reported by 85.42% of women in the IPV group and 10.42% in the non-IPV group. Women in the IPV group were more likely to have a psychiatric diagnosis, mainly depression and stress-related disorders. The detailed sociodemographic characteristics of the groups are presented in Table 1.

### 2.2. Measures

Demographics Form—The questionnaire was specifically designed to align with the study’s objectives and encompassed items pertaining to socioeconomic factors and experiences of violence.

The Young Schema Questionnaire—Short Form 3 (YSQ-SF3) was employed to assess the disconnection/rejection schema domain. To assess the D/R domain, the result obtained for 5 schemes (emotional deprivation, abandonment, mistrust/abuse, social isolation/alienation, defectiveness/shame) should be summed up. Each of the schemes is assessed by summing up 5 items, rated on a 6-point Likert scale (from 1 = completely untrue about me to 6 = describes me perfectly). The domain scores range from 25 to 150, with higher scores indicating greater D/R. [30]. In the Polish adaptation study, the questionnaire showed acceptable internal consistency, with overall Cronbach’s alpha score of 0.96 [43]. Cronbach’s alpha coefficient in our study indicates acceptable internal consistency (0.97).

The Resilience Measure Questionnaire (Kwestionariusz Oceny Prężności—KOP26) was used to assess the participant’s level of resilience. This questionnaire, created by Gasior, Chodkiewicz and Cechowski, includes 26 statements evaluated on a five-point Likert scale that refer to three components of resilience: family competence (org. Kompetencje Rodzinne—KR), personal competence (Kompetencje Osobiste—KO), social competence (Kompetencje Społeczne—KS). The original Polish version of the questionnaire was used. The internal consistency in the original study was excellent for total scale (Cronbach’s α = 0.90) and satisfactory for subscales (KR = 0.90, KO = 0.82, KS = 0.78) and correlates well with other resilience scales (i.e., The Ego Resiliency Scale in Kaczmarek’s adaptation ER/SPP, r = 0.59) [44]. The Cronbach’s alpha coefficient in our study indicates acceptable internal consistency (0.93).

The Parental Competence Test was used to measure parental competencies. The test consists of 30 short cases with descriptions of different upbringing situations, predominantly related to problematic behavior by the child. Each case is assigned three possible reactions of the parents; for each of them, the respondent determines the probability with which they would behave in a certain way. The assessment is made on a 4-point scale. The scale includes a total of 90 items. The test verifies five areas of parental behavior, the first one measuring parental competence, and the next four assessing incompetent parental behavior (rigorism, permissiveness, overprotectiveness, helplessness). The internal consistency for the normalization study was satisfactory for all subscales (from 0.68 to 0.91) [45]. In our study, the Cronbach’s alpha score indicates acceptable internal consistency (0.75).

The Parenting Attitude Scale was used to assess parental attitudes. The scale contains 50 items rated on a 5-point Likert scale that assess how strongly a parent agrees or disagrees with a diagnostic statement. The study used the version for mothers. This scale distinguishes five dimensions of parental attitudes: acceptance–rejection, autonomy, overprotectiveness, demands, and inconsistency. The first two of the abovementioned attitudes are considered favorable. The scale has high validity and reliability, with the internal consistency estimated with Cronbach’s α from female sample ranging from 0.74 (autonomy) to 0.89 (demands) [46]; in our study, it reaches 0.88. The Emotional Understanding Test (polish: Test Rozumienia Emocji—TRE) by Matczak and Piekarska was used to assess the ability to understand emotions, which is an important dimension of strategic emotional intelligence, which is the subject of interest in our research. The task-based formula aligns with the objectives of the study. The test consists of 30 items divided into 5 subtests, each of which contains 6 items. Each subtest focuses on a separate emotional comprehension task, including: (1) arranging emotional states based on their intensity; (2) recognizing the opposite emotion; (3) identifying the underlying emotion that contributes to the complex emotion; (4) matching emotional states to the described situations; (5) identifying the conditions that elicit specific emotional responses in specific situations. The sum of the points obtained in the subtests constitutes the total score (from 0 to 30). The reliability of the test estimated by Cronbach’s α was equal to or greater than 0.78 in the original study [47] and 0.70 for our study. The satisfactory methodological properties obatained encouraged us to choose this measurement.

### 2.3. Procedure

Recruitment for the study was carried out in cooperation with organizations providing multidimensional support for victims of domestic violence. We invited women to participate in the study during psychoeducational meetings and psychological workshops conducted in support institutions, ensuring the right to full anonymity and voluntariness. The inclusion criteria for the study were: separation from the perpetrator (intimate partner) for at least a month before the start of the study, stability of the participant’s mental state, which was assessed with the assistance of a qualified therapist (licensed psychologist, psychotherapist or crisis intervention specialist) and having full parental rights over at least one child. We recruited participants to the control group using social media, parenting portals and the snowball method. The study took place in a space designated for therapeutic work in the institutions we cooperated with under the constant assistantship of one of the researchers. Participants were informed about the possibility of dividing the study into two sessions. Babysitters were provided for children of the study participants throughout the test. Completing the tests set took an average of 90 min. Data collection lasted from April 2022 to March 2023. All participants provided signed informed consent to participate in the study. We obtained the ethical approval by the Bioethics Committee at the Medical University of Lodz (RNN/18/KE 12 June 2018) before data collection.

### 2.4. Statistical Analysis

Statistical analysis was performed using JASP 0.17.2.1. and STATISTICA 13.1 (TIBCO Software Inc., Palo Alto, CA, USA). Means, medians and standard deviations were calculated for continuous variables. Frequency counts (counts and percentages) were used to summarize the categorical variables. The normality of the distribution was verified using the Shapiro–Wilk test. Associations between the variables were tested by Spearman’s rank correlation coefficient. In the context of intergroup comparisons, the assumption of homogeneity of variance was not met. To address this, a non-parametric Mann–Whitney U test was employed.

Based on the results obtained, further exploratory analysis was conducted regarding the group of IPV victims (*n* = 48) to provide preliminary results that could provide direction for future research. While a larger sample size would be preferred for achieving satisfactory statistical power and generalizability, considering the exploratory character of this analysis, linear regression models were constructed and analysed in order to assess the predictive value of disconnection/rejection and possible mediation.

The goodness of fit of the model was assessed using the Fisher–Snedecor test. Through analysis of residuals, the validity of assumptions of normality, homoscedasticity and independence between observations (with the Durbin–Watson test) were assessed. To account for possible multicollinearities, tolerance indices were analysed, adopting lack of significant collinearity for a tolerance index greater than 0.1.

Two mediation models were created and analysed (Figure 1 and Figure 2). The first one, in which resilience (Resilience Measure Questionnaire) and strategic emotional intelligence (The Emotional Understanding Test) mediated the relationship between disconnection/rejection (YSQ-SF3) and autonomy (Parenting Attitude Scale) and the second one, in which resilience (Resilience Measure Questionnaire) mediated the relationship between disconnection/rejection (YSQ-SF3) and competence (Parental Competence Test). A post hoc analysis was performed to verify the statistical power. Bootstrapping, with sampling set at *n* = 5000, was performed to empower the results and account for a possible nonparametric distribution.

Statistical significance was defined as *p* < 0.05 or a confidence interval not encompassing 0.

## 3. Results

### 3.1. Differences in Parenting Practice, Levels of Resilience and Strategic Emotional Intelligence and Severity of Disconnection and Rejection Schema Domain between Women with and without IPV History

The characteristics of the measures, including their means, standard deviations and the results of intergroup comparisons conducted using the U value with the corresponding level of statistical significance are shown in Table 2. Significant differences were observed between the research group and the reference group with regard to the studied variables. In terms of parenting practice, women who were victims of IPV scored significantly lower on the scales of parental competence and permissiveness, and significantly higher on scales of overprotectiveness, demands and rigorism. Another notable difference relates to the severity of the D/R schema domain, where women with IPV experience scored significantly lower in terms of personal resources scales, specifically resilience and strategic emotional intelligence, than women from the reference group.

### 3.2. The Relationship between Parenting Practice, Resilience and Strategic Emotional Intelligence and Schemas of the Disconnection and Rejection Domain

As shown in Table 3, statistically significant correlations were found among the continuous variables examined in the study. These correlations indicated negative and weak to moderate associations between unfavorable parenting practice and strategic emotional intelligence, as well as resilience. Additionally, positive and weak to moderate associations were found between unfavorable parenting practice and the D/R domain. Further, the analysis revealed negative and weak correlations between the D/R domain and both parental competence and permissiveness.

### 3.3. Resilience and Strategic Emotional Intelligence as Mediators in the Relationship between the Disconnection and Rejection Schema Domain and Parenting Practice

The proprieties of models predicting autonomy (Parenting Attitude Scale) and competence (Parental Competence Test) are presented in Table 4 and Table 5, respectively. For each model, the possible mediation effects were tested by analyzing the direct and indirect effects presented in Table 6 and Table 7.

## 4. Discussion

Overall, this study aimed to investigate differences in parenting practice between women with and without a history of IPV. Furthermore, we sought to provide preliminary results exploring whether the D/R schema domain could be a significant predictor of unfavorable parenting practices and whether resilience and strategic emotional intelligence could act as mediators to mitigate this relationship. The results revealed that women who experienced IPV were more likely to use unfavorable parenting practices than women in the reference group. These findings align with previous research, further supporting the assumption that IPV experience negatively affects parental behavior. First, as reported previously [48], in our study, women who experienced IPV scored significantly higher on overprotectiveness, which was consistently reflected in the data from both measures of parenting practice. Although the underlying motive for overprotection is often a desire to keep the child safe, an overemphasis on preventing harm ultimately limits a child’s chances of nurturing a sense of healthy autonomy and developing the necessary self-regulation and psychosocial skills [49]. Meta-analyses have confirmed the detrimental effects of overprotective parenting on individuals at different stages of life. In childhood, overprotectiveness has been linked to internalizing problems, depression and anxiety [50]. Similarly, in adulthood, overprotective parenting was associated with mood disorders, anxiety and related mental disorders [51]. Additionally, women who experienced IPV scored significantly lower on a parenting competence scale. These findings are consistent with a meta-analysis examining the relationship between IPV victimization and parenting, which revealed a significant, albeit small, negative correlation between IPV experience and positive parenting. That mothers with IPV experience, through different pathways, are less able to parent their children consistently, develop self-knowledge and support self-regulation skills, was also considered as mediator between IPV exposure and children’s health outcomes [20]. Finally, similarly to Greene et al. [22], we found that IPV victims scored significantly higher on the rigorism and demands scales, thus employing more restrictive and punitive parenting strategies with their children. At this level, our findings indicate that IPV may negatively affect maternal parenting, as hypothesized by the spillover theory [16].

As expected, the D/R schema domain was found to be significantly associated with unfavorable parenting behaviors. Repeatedly, this domain has been identified as the most deleterious in its impact [52,53,54]. Although the majority of studies have retrospectively investigated the relationship between parenting practices and subsequent development of EMS [55], only a few have examined the impact of EMS on parenting [32]. Given this, the potential for a direct comparison of our findings in this regard is limited. Nevertheless, the D/R domain has been reported to adversely affect parenting. For example, in the study by Nordahl et al., it was observed that the D/R domain emerged as the most accurate predictor of prenatal bond quality, showing a negative correlation [56]. In terms of schema theory, the domain of D/R results from the deprivation of basic emotional needs [30] and is also closely connected with the experience of neglect and abuse in the family environment [31]. In regard to this association, our findings parallel studies showing links between a history of child abuse and neglect and negative parenting in adulthood [57,58,59]. Another explanatory mechanism for the examined relationship is the detrimental impact of the D/R domain schemas on mental well-being [60,61]. In this context, the results align with previous research indicating that the mother’s mental health issues may significantly impair parenting abilities [22].

Furthermore, the present study initiates insight into the role of personal resources that could mitigate the associations between the D/R domain and unfavorable parenting practices among IPV victims. The analysis showed a significant direct effect between the D/R domain and the parental attitude of autonomy, as well as between the D/R domain and the parental competence scale. The introduction of resilience with strategic emotional intelligence and resilience into the models, respectively, weakened the strength of the aforementioned effects to a level below statistical significance, which indicates that it is in a state of full mediation. In our study, resilience refers to three sets of protective resources: family competencies, personal competencies and social competencies. Although the concept of parental resilience is still underdeveloped, the research to date has indicated that family and social functioning, as well as self-efficacy, play an important role in parents’ ability to meet the multiple challenges of raising children [37,62]. This observation is also emphasized in our research. Finally, strategic emotional intelligence turned out to be a significant predictor of parental autonomy attitude but did not predict the level of parental competence. This discrepancy may be due to the fact that the parental competence scale measures the complex cognitive and emotional abilities of parents aimed at supporting the child’s self-regulatory skills, whereas strategic emotional intelligence refers to understanding emotions at a basic level. Employing multidimensional measurement tools to establish the relationship between emotional intelligence and parenting practice is strongly recommended for further research. Nevertheless, strategic emotional intelligence facilitates the attitude of parental autonomy, also acting as a mitigating factor for the negative relationship between the D/R domain and the attitude of parental autonomy, which is generally consistent with previous research [41,42].

Implications for clinical practice and further research can be drawn based on the study findings. EMS assessment can help identify IPV victims at higher risk of parenting difficulties. This can be crucial not only because of the possibility of implementing preventive measures aimed at improving the mother–child relationship but also due to treatment matters. As previously established, schema therapy is an effective method of alleviating EMS severity and improving the overall mental health condition [63]. As we have identified factors that could mitigate the devastating impact of the D/R domain on the mother–child dyad in the context of IPV, approaches that enhance resilience and strategic emotional intelligence are strongly recommended. Such interventions can go beyond the individual level and include mother–child interaction in a comprehensive manner, e.g., by inducing positive affect in the mother–child relationship through cooperative motor games [64]. Further research exploring the relationship between EMS and parental practices is necessary not only in the context of IPV. We encourage the duplication of research with other, preferably larger samples, including clinical and non-clinical populations, and expanding the range of measurements with more comprehensive tools that explore personal resources.

The study has some limitations that should be mentioned. First, the sample size used in this study was relatively small, which limits the power of statistical analyses. As a consequence, when considering the results, their preliminary character should not be disregarded. Hence, it is highly recommended that future research comprises larger sample sizes in order to verify our findings and improve the statistical power and conclusiveness of the results. Another potential limitation is the use of self-report methods, which, especially when evaluating parental attitudes, may introduce biases that may distort the results. Levendosky’s research revealed discrepancies between maternal self-reports and observational measurements [24]. Therefore, future studies may increase the validity of the findings by using a combination of different assessment methods. Lastly, it is worth mentioning that the compared groups in this study were not homogeneous with regard to certain demographic variables, potentially influencing the nature of the observed differences. A valuable approach for future research could involve closely matching the reference group in terms of sociodemographic factors to mitigate potential distortions in the findings.

Taken together, in our research we employed mediation models to investigate how resilience and strategic emotional intelligence contribute to the connection between women’s D/R schema domain and their maternal parenting practices within a group of women who experienced IPV victimization. We anticipated that: (1) Women who experienced IPV would score higher on scales measuring adverse parenting practices and lower scores on scales measuring parenting competence. (2) The severity of the D/R domain will be related to adverse parenting practices and parental competence, with positive and negative correlations, respectively. (3) Resilience and strategic emotional intelligence will mediate the relationship between the mother’s D/R schema domain and her adverse parenting practices as well as parenting competence. These hypotheses were reflected in the preliminary results obtained in our study. This study showed that IPV victimization is associated with significantly higher rates of negative parenting practices, adding to the evidence supporting spillover hypothesis. The D/R domain was found to likely be a significant predictor of parental autonomy attitude and the level of parental competence, and these relationships seem to be fully mediated by resilience with strategic emotional intelligence and resilience, respectively. Our results shed light on the under-researched relationship between EMS and parenting behavior, especially in the context of IPV. They could serve as preliminary results for larger sample studies and encourage greater interest in this area. In the future, strategies combining schema therapy intervention with enhancing resilience and emotional intelligence are strongly recommended.

## Figures and Tables

**Figure 1 brainsci-13-01290-f001:**
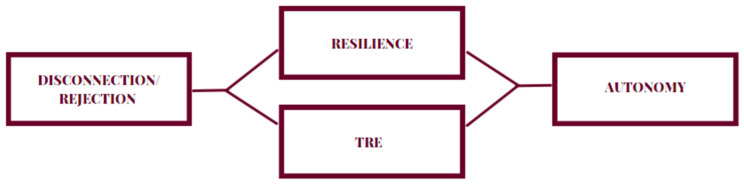
The hypothesized model of the mediating role of resilience and strategic emotional intelligence in the relationship between the disconnection and rejection schema domain and autonomy (measured by the Parenting Attitude Scale).

**Figure 2 brainsci-13-01290-f002:**
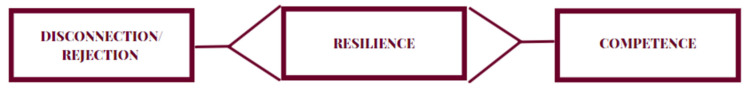
The hypothesized model of the mediating role of resilience in the relationship between the disconnection and rejection schema domain and competence (measured by the Parental Competence Test).

**Table 1 brainsci-13-01290-t001:** Sociodemographic characteristics of participants.

Variables	IPV * Group	Non IPV * Group
	*n* (%)	*n* (%)
Age		
M ± SD	32.91 ± 7.79	34.27 ± 2.90
Range	20–55	28–42
Education		
Primary	18 (37.5)	-
Vocational	6 (12.5)	-
Secondary	13 (27.08)	6 (12.5)
Higher	11 (22.92)	42 (87.5)
Marital status		
Marriage	17 (35.42)	39 (81.25)
Informal relationship	9 (18.75)	7 (14.58)
Divorced	5 (10.42)	2 (4.17)
No relationship	17 (35.42)	-
Place of residence		
Countryside	2 (4.17)	14 (29.17)
City up to 50,000 residents	4 (8.33)	4 (8.33)
City 50,000–100,000 residents	11 (22.92)	3 (6.25)
City over 100,000 residents	31 (64.58)	27 (56.25)
Children		
1	18 (37.5)	27 (56.25)
2	16 (33.33)	20 (41.67)
3	14 (29.17)	1 (2.08)
Childhood victimatization	28 (58.33)	1 (2.08)
Witnessing Violence in childhood	41 (85.42)	5 (10.42)
Psychiatric diagnosis	23 (47.92)	9 (18.75)

* IPV = intimate partner violence.

**Table 2 brainsci-13-01290-t002:** Comparison of the measures.

Measures	Non-IPV Group			IPV Group			U	Z	*p*
	Mean	St.Dev	Rank Sum	Mean	St.Dev	Rank Sum			
Emotional Understanding ^1^									
Sten scores	5.81	1.75	2902.0	3.79	2.32	1754.0	**578.0**	**4.20**	**0.000**
Resilience ^2^									
Total score	104.45	16.83	2842.0	89.76	18.74	1814.0	**638.0**	**3.76**	**0.000**
Family Competence	47.78	7.98	2973.0	38.06	9.95	1683.0	**507.0**	**4.72**	**0.000**
Personal Competence	37.11	6.47	2688.0	33.57	6.76	1968.0	**792.0**	**2.63**	**0.008**
Social Competence	19.58	5.23	2482.0	18.20	5.91	2174.0	998.0	1.12	0.264
Parenting Attitude ^3^									
Acceptance–Rejection	45.29	4.00	2083.0	45.98	5.20	2573.0	907.0	−1.792	0.073
Autonomy	40.25	4.56	2545.0	38.23	6.49	2111.0	935.0	1.586	0.112
Overprotectiveness	24.52	7.12	1553.0	37.02	9.95	3103.0	**377.0**	**−5.675**	**0.000**
Demands	24.25	6.58	1844.5	30.77	9.12	2811.5	**668.5**	**−3.539**	**0.000**
Inconsistency	21.23	7.16	2122.0	23.63	7.94	2534.0	946.0	−1.506	0.132
Parental Competence ^4^									
Competence	131.52	7.62	2754.0	127.0	8.72	1902.0	**726.0**	**3.118**	**0.002**
Rigorism	52.86	7.98	1686.5	63.33	11.34	2969.5	**510.5**	**−4.700**	**0.007**
Permissiveness	36.38	5.21	2695.0	33.00	6.07	1961.0	**785.0**	**2.686**	**0.010**
Overprotectiveness	56.17	7.69	1545.0	69.48	10.69	3111.0	**369.0**	**−5.734**	**0.000**
Helplessness	34.67	5.64	2494.5	33.40	5.51	2161.5	985.5	1.216	0.224
Dissconnection/Rejection ^5^	52.17	20.09	1887.5	81.55	24.23	3572.5	**456.5**	**−5.82**	**0.000**

^1^ The Emotional Understanding Test (pol. Test Rozumienia Emocji (TRE); ^2^ Resilience Measure Questionnaire (pol. Kwestionariusz Oceny Prężności—KOP26); ^3^ Parenting Attitude Scale; ^4^ Parental Competence Test; ^5^ YSQ-SF3 = Young Schema Questionnaire—Short Form 3. Marked tests are significant at *p* < 0.05.

**Table 3 brainsci-13-01290-t003:** Correlation matrix for tested variables.

Variable	1	2	3	4	5	6	7	8	9	10	11	12	13	14	15	16	17	18
A/R ^1^	1.00																	
Aut ^2^	**0.22**	1.00																
Ov-p ^3^	**0.29**	−0.18	1.00															
Dem ^4^	−0.03	−0.19	**0.61**	1.00														
Inc ^5^	−0.16	−0.13	**0.32**	**0.44**	1.00													
Comp ^6^	**0.28**	0.03	−0.06	−0.04	−0.05	1.00												
Rig ^7^	0.08	−0.14	**0.46**	**0.55**	**0.30**	−0.15	1.00											
Perm ^8^	0.08	0.10	**−0.32**	**−0.22**	0.02	**0.26**	−0.12	1.00										
Ov-p ^9^	**0.26**	−0.14	**0.67**	**0.46**	**0.22**	−0.12	**0.65**	−0.18	1.00									
Help ^10^	−0.04	0.12	**−0.26**	**−0.28**	−0.10	**−0.33**	−0.08	0.04	0.05	1.00								
TRE ^11^	−0.17	**0.22**	**−0.53**	**−0.51**	−0.13	0.08	**−0.49**	0.13	**−0.54**	0.09	1.00							
Ed ^12^	0.10	−0.07	**0.49**	**0.26**	0.14	**−0.31**	**0.42**	**−0.23**	**0.47**	−0.11	**−0.29**	1.00						
Ab ^13^	0.02	**−0.22**	**0.34**	**0.27**	0.16	−0.06	**0.28**	−0.12	**0.38**	−0.16	−0.14	**0.58**	1.00					
M/A ^14^	0.15	−0.13	**0.48**	**0.33**	**0.23**	−0.12	**0.39**	**−0.25**	**0.43**	−0.20	**−0.28**	**0.66**	**0.69**	1.00				
Si ^15^	0.14	0.04	**0.26**	0.04	0.06	**−0.23**	0.20	−0.11	**0.23**	−0.04	−0.05	**0.59**	**0.50**	**0.62**	1.00			
D/S ^16^	−0.02	−0.12	**0.40**	0.12	0.14	**−0.22**	**0.30**	−0.09	**0.42**	−0.08	−0.19	**0.69**	**0.65**	**0.59**	**0.70**	1.00		
Res ^17^	0.04	0.02	−0.09	−0.07	−0.12	0.12	−0.18	0.05	**−0.22**	−0.01	0.09	**−0.38**	**−0.27**	**−0.36**	**−0.32**	**−0.26**	1.00	
D/R	−0.03	−0.04	**0.38**	**0.33**	**0.22**	**−0.26**	**0.37**	**−0.21**	**0.44**	−0.06	**−0.26**	**0.55**	**0.55**	**0.63**	**0.50**	**0.55**	**−0.32**	1.00

Parenting Attitude Scale: ^1^ Acceptance–Rejection; ^2^ Autonomy; ^3^ Overprotectiveness; ^4^ Demands; ^5^ Inconsistency; Parental Competence Test: ^6^ Competence; ^7^ Rigorism; ^8^ Permissiveness; ^9^ Overprotectiveness; ^10^ Helplessness; The Emotional Understanding Test (pol. Test Rozumienia Emocji): ^11^ TRE Sten score; Young Schema Questionnaire—Short Form 3; ^12^ Emotional deprivation; ^13^ Abandonment; ^14^ Mistrust/Abuse; ^15^ Social Isolation/Alienation; ^16^ Defectiveness/Shame; Resilience Measure Questionnaire (pol. Kwestionariusz Oceny Prężności—KOP26): ^17^ Resilience. Marked correlations are significant at *p* < 0.05.

**Table 4 brainsci-13-01290-t004:** A summary of the hierarchical linear regression analyses predicting autonomy (Parenting Attitude Scale) among IPV victims (*n* = 48). The data are presented as unstandardized parameter (β) and bias corrected accelerated 95% confidence intervals (BCa95% CI). Results are derived by bootstrapping with *n* = 5000 sampling.

Disconnection/Rejection ^1^ Domain as Autonomy ^2^ Predictor (R^2^ = 0.058; F = 6.269; df = 1; *p* = 0.014)				
Predictor	β	βCa95%CI	t	*p*
Disconnection/Rejection ^1^	−0.052	[−0.092; −0.013]	−2.504	0.008
Full model (R^2^ = 0.151; F = 5.927; df = 3; *p* < 0.001)				
Predictor	β	βCa95%CI	t	*p*
Disconnection/Rejection ^1^	−0.010	[−0.059; 0.039]	−0.391	0.694
Resilience ^3^	0.072	[−0.005; 0.139]	2.143	0.035
TRE ^4^	0.315	[−0.065; 0.565]	2.499	0.014

R^2^—coefficient of determination; F—Fisher–Snedecor test statistics; df—degrees of freedom; *p*—probability in the test; ^1^ YSQ-S3—Young Schema Questionnaire-L3; ^2^ Parenting Attitude Scale; ^3^ Resilience Measure Questionnaire (pol. Kwestionariusz Oceny Prężności—KOP-26; ^4^ The Emotional Understanding Test (pol. Test Rozumienia Emocji).

**Table 5 brainsci-13-01290-t005:** A summary of the hierarchical linear regression analyses predicting competence (Parental Competence Test) among IPV victims (*n* = 48). The data are presented as unstandardized parameter (β) and bias corrected accelerated 95% confidence intervals (BCa95% CI). Results are derived by bootstrapping with *n* = 5000 sampling.

Disconnection/Rejection ^1^ Domain as Competence ^2^ Predictor (R^2^ = 0.055; F = 5.966; df = 1; *p* = 0.016)				
Predictor	β	βCa95%CI	t	*p*
Disconnection/Rejection ^1^	−0.076	[−0.137; −0.016]	−2.443	0.013
Full model (R^2^ = 0.107; F = 4.015; df = 3; *p* < 0.010)				
Predictor	β	βCa95%CI	t	*p*
Disconnection/Rejection ^1^	−0.022	[−0.098; 0.054]	−0.574	0.567
Resilience ^3^	0.125	[−0.021; 0.229]	2.375	0.019
TRE ^4^	0.085	[−0.304; 0.474]	0.435	0.665

R^2^—coefficient of determination; F—Fisher–Snedecor test statistics; df—degrees of freedom; *p*—probability in the test; ^1^ YSQ-S3—Young Schema Questionnaire-L3; ^2^ Parental Competence Test; ^3^ Resilience Measure Questionnaire (pol. Kwestionariusz Oceny Prężności—KOP-26; ^4^ The Emotional Understanding Test (pol. Test Rozumienia Emocji).

**Table 6 brainsci-13-01290-t006:** The direct and indirect (i.e., mediated by resilience and TRE) effect of disconnection/rejection (YSQ-S3) on autonomy (SPR) among IVP victims (*n* = 48). The data are presented as standardized estimates from linear regression models, with 95% confidence intervals (CI), derived by bootstrapping with *n* = 5000 sampling.

Direct Effect of Disconnection/Rejection ^1^ on Autonomy ^2^	Estimate	95%CI
Disconnection/Rejection ^1^	−0.010	[−0.062; 0.041]
Indirect Effect of Disconnection/Rejection ^1^ on Autonomy ^2^	Estimate	95%CI
Total Indirect Effects	−0.042	[−0.090; −0.011]
Resilience ^3^	−0.030	[−0.059; 0.001]
TRE ^4^	−0.012	[−0.032; −0.002]

^1^ YSQ-S3—Young Schema Questionnaire-L3; ^2^ Parenting Attitude Scale; ^3^ Resilience Measure Questionnaire (pol. Kwestionariusz Oceny Prężności—KOP-26; ^4^ The Emotional Understanding Test (pol. Test Rozumienia Emocji).

**Table 7 brainsci-13-01290-t007:** The direct and indirect (i.e., mediated by resilience) effect of disconnection/rejection (YSQ-S3) on competence (Parental Competence Test) among IPV victims (*n* = 48). The data are presented as standardized estimates from linear regression models, with 95% confidence intervals (CI), derived by bootstrapping with *n* = 5000 sampling.

Direct Effect of Disconnection/Rejection ^1^ on Competence ^2^	Estimate	95%CI
Disconnection/Rejection ^1^	−0.025	[−00.103; 0.048]
Indirect Effect of Disconnection/Rejection ^1^ on Competence ^2^	Estimate	95%CI
Total Indirect Effects	−0.077	[−00.134; −00.014]
Resilience ^3^	−0.051	[−00.130; −00.008]

^1^ YSQ-S3—Young Schema Questionnaire-L3; ^2^ Parental Competence Test; ^3^ Resilience Measure Questionnaire (pol. Kwestionariusz Oceny Prężności—KOP-26.

## Data Availability

The data presented in this study are available on request from the corresponding author. The data are not publicly available due to privacy issues.

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
