# Peer review of "Resilience and Strategic Emotional Intelligence as Mediators between the Disconnection and Rejection Domain and Negative Parenting among Female Intimate Partner Violence Victims"

_brainsci, 2023, doi:10.3390/brainsci13091290_

Round 1

Reviewer 1 Report

The subject that this work deals with, such as resilience, emotional intelligence and parenting in women victims of intimate partner violence, seems to me of great interest.

I think that the summary must be reformulated, it must be modified. Indicate the objective, the method (context, participants, measurement instruments, analysis) and results or conclusions.

I believe that in the participants section, the authors should establish the geographical context of the study, as this is key to understanding the study.

I think it should be better explained how the participants were recruited, the procedure for selecting the participants

I think that in the procedure section it should be explained when and where the participants filled out the instruments used, the location, the confidence that was given to them, as well as whether this was done in a single session or in several sessions given the large number of items. to answer

I think the authors should justify why they use the instruments they use to measure emotional understanding and not others or use an adjective scale or other affective domain tests.

To this end, I ask the authors to review these works, as they will be of great interest in the discussions of their study, since they are studies that have measured emotions in different contexts.

Gil-Madrona, P., Pascual-Francés, L., Jordá-Espi, A., Mujica-Johnson, F., & Fernández-Revelles, A. B. (2020). Affectivity and Motor Interaction in Popular Motor Games at School. notes. Physical Education and Sports, 139, 42-48. https://doi.org/10.5672/apunts.2014-0983.es.(2020/1).139.06

analytics are ok

Table presentation is fine

discussions are fine

Reviewer 2 Report

Thanks to the authors for sharing their manuscript. I congratulate them on their interesting and scientifically valuable research. I would like to mention two points that cause me concern.

Firstly, the authors describe the measures in great detail, note the Polish adaptations and even the psychometric properties of the adapted versions of the measures, but do not indicate the Cronbach’s alphas in their study. It is very important to specify the Cronbach's alphas so that it can be seen that the measures can be used for this sample.

Secondly, the sample is very complex, which, of course, explains its small size. In the description of the limitations, the authors refer to the fact that a relatively small sample limits the power of statistical analyses, but they have not previously given a calculation of the sample size and statistical power. This raises the question: was it possible to conduct regression analysis and mediation analysis?

If the authors calculate the Cronbach’s alphas and statistical power and find acceptable values, then I recommend that these data be included in the manuscript. If some indicators are unacceptable, it will be necessary to expand the sample or use other types of statistical analysis.
